# A Malicious Code Detection Method Based on FF-MICNN in the Internet of Things

**DOI:** 10.3390/s22228739

**Published:** 2022-11-12

**Authors:** Wenbo Zhang, Yongxin Feng, Guangjie Han, Hongbo Zhu, Xiaobo Tan

**Affiliations:** 1School of Information Science and Engineering, Shenyang Ligong University, Shenyang 110159, China; 2Department of Information and Communication Systems, Hohai University, Changzhou 213022, China

**Keywords:** IoT, malicious code detection, classification detection of images, improved convolutional neural network, FF-MICNN

## Abstract

It is critical to detect malicious code for the security of the Internet of Things (IoT). Therefore, this work proposes a malicious code detection algorithm based on the novel feature fusion–malware image convolutional neural network (FF-MICNN). This method combines a feature fusion algorithm with deep learning. First, the malicious code is transformed into grayscale image features by image technology, after which the opcode sequence features of the malicious code are extracted by the n-gram technique, and the global and local features are fused by feature fusion technology. The fused features are input into FF-MICNN for training, and an appropriate classifier is selected for detection. The results of experiments show that the proposed algorithm exhibits improvements in its detection speed, the comprehensiveness of features, and accuracy as compared with other algorithms. The accuracy rate of the proposed algorithm is also 0.2% better than that of a detection algorithm based on a single feature.

## 1. Introduction

With the rapid development of scientific and technological information, many new core technologies have been applied in various fields. In particular, the mutual integration and innovation of artificial intelligence (AI), 5G, and the Internet of Things (IoT) promote the continuous expansion of the scale of the IoT industry, which not only has a positive influence on the industrial field, but also a significant impact on daily life [1].

In the IoT environment, sensor devices are connected through the internet protocol and they exchange information and communicate through the medium of information transmission. Due to the communication characteristics of IoT, the number of attacks, such as distributed denial of service (DDoS), buffer overflow, botnet, malicious code encryption, and ARP spoofing attacks is rapidly expanding [2]. In addition, the development of IoT technology has led to increasing demand for IoT devices. To meet this demand, many manufacturers produce devices with low security and sensitivity to vulnerabilities, thus making these devices targets of attack and further aggravating the possibility of malicious attacks. According to research by IBM, the number of internet-connected devices will increase to 50 billion by 2020. The Kaspersky Lab collected 121,588 IoT malicious code samples in 2018, about four times more than the 32,614 samples it collected in 2017. Among these samples, more than 120,000 samples of mutated IoT malicious code were found, and the attack methods were intelligently evolved [3]. To prevent IoT devices from being attacked by new or transformable malicious code, it is meaningful to detect malicious code [4].

In view of these factors, our motivation is to overcome the shortcomings of existing algorithms for malicious code detection. Aiming at the static and multi-layer features of malicious code detection, we would use the method of image and feature fusion to design the detection algorithm, so as to improve the detection speed, the comprehensiveness of features, and the accuracy. Therefore, a malicious code detection algorithm based on FF-MICNN is proposed in the present study. The main contributions of this paper are as follows:A grayscale image converted from malicious code is used as the input for the improved network model, and the malicious code detection task is converted into an image classification task;The FF-MICNN algorithm is proposed. The opcode sequence features and grayscale image features are fused, and the improved CNN is used for detection.

The remainder of this article is organized as follows. Section 2 describes the previous work related to malicious code detection methods. Section 3 introduces the proposed FF-MICNN algorithm in detail. The simulation results and performance analysis are discussed in Section 4. Finally, Section 5 presents the conclusion and describes directions for future research.

## 2. Related Work

Many researchers have developed different algorithms and solutions to address the research questions discussed in the introduction. The existing methods of malicious code detection mainly include static detection algorithms [5,6,7] and dynamic detection algorithms [8,9].

The static detection method is to identify malicious code by analyzing the data at the data level and capturing the relevant semantic and grammatical information of the data without running the malicious code. Many scholars have carried out numerous studies on different data structures and forms as shown in Table 1.

Due to the deficiency of static detection methods, dynamic detection methods have been developed as shown in Table 2. This type of detection method generates behavior reports for portable executable files by analyzing the execution code in a virtual environment and is based on the execution tracking of the code.

Image detection methods are different from traditional detection methods. This type of method is improved on the basis of static detection methods via a relatively new approach to analyze the binary files of malware, and it also detects confused malware, new malware, and malware variants by converting the malware into image features. The concept of malicious code visualization has been proposed and is widely used, and realizes malicious code detection by converting malicious code into a grayscale image and further processing the image [23]. The typical image detection methods is shown in Table 3.

Based on the preceding literature review, the present work proposes a malicious code detection method based on the FF-MICNN to overcome the deficiency of the traditional detection of malicious code varieties and the use of single features. The proposed method combines both feature fusion and deep learning algorithms. First, malicious code is transformed into grayscale image features by visualization technology. Then, the opcode sequence features of the malicious code are extracted by the n-gram technique. Finally, the global and local features are fused by feature fusion technology, the obtained fusion features are input into FF-MICNN for training, and the appropriate classifier is selected for detection.

## 3. FF-MICNN Algorithm

With the rapid development of the IoT, many devices are connected to the Internet. Deep learning [28,29,30,31] has been applied to many fields, and its effects and achievements have attracted attention. Among the various deep learning frameworks, the CNN is the most outstanding in image processing as compared with other networks. The algorithm proposed in this paper is a network model algorithm adapted to the malicious code environment based on the improvement of the CNN.

Figure 1 demonstrates the overall composition of the malicious code detection method based on FF-MICNN, which mainly includes three parts: data processing, model training, and classification detection. The data processing component mainly includes two processes, namely feature extraction and feature fusion. The purpose of feature extraction is to obtain the opcode sequence features and grayscale image features in the dataset with a .asm file suffix. In the feature fusion component, two extracted single features are fused to form a new feature vector, which is used as the input of the deep learning model and the training model. During model training, the parameters of the model are set and optimized by constructing the FF-MICNN, and a set of parameter combinations suitable for the training of the network model is selected so that the network can be self-adapted for classification detection. The classification detection component uses classifiers to classify malicious codes into their corresponding categories.

### 3.1. Data Processing

#### 3.1.1. Grayscale Image Feature Extraction

The malicious code visualization method adopts image processing technology to convert malicious code files into corresponding grayscale images, which can be used to further analyze malicious code. There is no need to execute the code to convert the malicious code into a grayscale image. Compared with the manual analysis of confused malicious code, this not only saves time and reduces the analysis of the code, but also avoids the harm caused to computers by malicious code. The conversion process of this method is shown in Figure 2, and the specific conversion process is as follows.

First, the decompiled .asm files in the dataset are read in binary form in units of 8 bits. Then, the binary sequence of each unit is converted into an unsigned decimal numeric form. The decimal range is between 0 and 255, and different values represent any pixel value in the image; 0 is black, 255 is white, and other values are in between black and white. Finally, the resulting decimal value is converted into a two-dimensional array, and the process is repeated until the binary file is fully read.

#### 3.1.2. Feature Extraction of Opcode Sequences

The opcode is located in the .text code section of the malicious code file, which describes the relevant instruction behavior of the malicious code. It can most accurately describe the local characteristics of the malicious code. The realization of the extraction of opcode sequence features is as follows.

First, the opcodes are stored in the .text code section in the .asm file, and the contents of this section must be read as the line reads. The read content is then converted in hexadecimal fashion. Next, the regular expression is used to match what is read on each line, which either contains a complete operating instruction or an instruction containing opcodes and operations. Finally, the opcodes are extracted from the matched instructions until all the opcodes of the file are extracted and the opcode sequence of the file is obtained. This algorithm is presented in Algorithm 1.
**Algorithm 1** Extraction algorithm of opcode sequences**Input:** 
.asm file;**Output:** 
Opcode sequence;1:Defines a variable to store an opcode sequence2:Regularization matching3:Open the .asm file4:Read the file by line5:Judgment, start reading with the .text section6:Read the content7:Judgment, match to the content8:Match, extract opcode; Match failure, this content is followed by an end opcode identifier9:Return the resulting opcode sequence10:Finish

During feature extraction, if the extracted opcode sequence is too long, the model training cannot achieve the ideal effect. Therefore, the *n*-gram algorithm is used to display the opcode sequence in the form of a feature vector space. The basic idea is to manipulate the content information in the text according to the size of the byte *n*-value by using the sliding window method. The sequence of byte fragments with the same length of *n* is realized, and the frequency of the occurrence of all obtained grams with length *n* is counted. Then, filtering is carried out according to the set threshold, the grams that do not meet the threshold are deleted, and the gram list is formed as a reference  [32]. After testing, it was found that the detection effect was the best when the value of *n* was 3 and the occurrence frequency was 700. Therefore, in the experiment conducted in this study, the opcode sequence features of n=3 and occurrence frequency = 700 were selected.

#### 3.1.3. Feature Fusion

At the local and global levels, the opcode sequence and grayscale image features of malicious code represent the similarity of malicious code [33]. There will be some disadvantages if the similarity of malicious code is represented only by local opcodes [34]. Although it cannot completely describe the similarity of malicious code, it can represent the features of the core code segment to some extent. The grayscale image can represent not only the features of the core code segment of malicious code but also the features of other data resource segments; however, it cannot obtain enough local feature information [35]. During the training process of the deep learning network, regardless of whether local or global features are used, there may be incomplete feature expression. Therefore, in this study, the local features of opcodes are combined with the global features of grayscale images, based on which a high-performance malicious code classification and detection method is achieved. The core algorithm of the feature fusion method used in this study is shown in Algorithm 2.
**Algorithm 2** Feature fusion algorithm**Input:** 
Opcode sequence feature f1 and grayscale image feature f2;**Output:** 
Feature *f* after feature fusion;1:Read the first opcode sequence feature f12:Read the second grayscale image feature f23:Read a label file with malicious code labels4:Find the two features according to the ID of the malicious code5:Superimpose the opcode sequence feature vectors on the end of the feature vectors of the grayscale image and fuse them by the function called pandas.merge()6:Find the corresponding tag based on the ID of the malicious code and merge the two tags with the function called panda.merge()7:Obtain the tagged feature *f* after feature fusion8:Finish

### 3.2. Model Building

Compared with machine learning and backpropagation (BP) methods, the advantage of the CNN is the reduction of the number of parameters of the network training process via local connection and weight sharing [35,36]. However, a CNN requires that the input image size be fixed. To solve this problem, the FF-MICNN is proposed for the feature extraction and detection of malicious codes. It cannot only extract features by self-learning but also reduces the model parameters and amount of computation while ensuring accuracy [37]. This model can realize the feature extraction and detection of images of different sizes, which is an improvement based on the CNN, and its composition is demonstrated in Figure 3.

First, the input of the input layer is the fused features. Then, the convolution layer extracts the input features. The weight-sharing function of the network cannot only reduce the network parameters and retain the main features of the grayscale image but can also reduce the influence of noise. Then, the pooling layer is located behind the convolution layer. It selects the features of the feature graph output by the convolution layer, filters out the irrelevant information, realizes the dimensionality reduction of the data, and carries out multiple convolutional pooling processes to obtain the most effective features. Next, the added layer can realize the output of a fixed feature number by selecting different pooling windows. The fully connected layer can then incorporate local features that increase the output of the layer. Finally, the integration results are input into the softmax layer for category judgment to realize the detection of malicious code. The detailed iteration formula for each layer is as follows.

#### 3.2.1. Convolution Layer

The convolution layer is the first layer in which input data are processed. The main function is to extract the characteristics of the input grayscale image. The weight-sharing function of the network can reduce the network parameters, retain the main features of the grayscale image, and reduce the influence of noise. Each neuron in the convolution layer is connected with the coefficient of the convolution value output by the upper layer, and the operation calculation is as follows:(1)$$x_{j}^{l}=f\left(\sum_{i\in M_{j}}x_{j}^{l-1}*k_{ij}^{l}+b_{j}^{l}\right),$$
where $M_{j}$ is the input feature mapping set, $k_{ij}^{l}$ is the connection of the core of the *i*-th input property and the weight of the *j*-th output feature map, and $b_{j}^{l}$ is the offset corresponding to the *j*-th feature mapping.

#### 3.2.2. Pooling Layer

The pooling layer processes the output of the convolution layer. The main function is to carry out feature selection based on the output of the convolutional layer, which cannot only filter out irrelevant information and realize the dimensionality reduction of data, but can also reduce the influence on image deformation and the dimension of image features, thus improving the accuracy of the model [38]. The operation calculation of the pooling layer is
(2)$$x_{j}^{l}=f\left(down\left(x_{j}^{l-1}\right)+b_{j}^{l}\right),$$
where down(.) represents a sub-sampling function and $b_{j}^{l}$ is the deviation.

#### 3.2.3. Added Layer

This layer is located before the fully connected layer and after the last pooling layer. First, the input criteria for the CNN are fixed. Second, the disassembly file of malicious code is caused by the different sizes of information storage, and the converted grayscale images are also different sizes. Hence, this cannot meet the input criteria of the network model. However, the input criteria for the network model are determined by the fully connected layer. Neurons in the fully connected layer are fixed and are fully connected to the neurons in the previous layer. Therefore, as long as the sizes of the grayscale image features are guaranteed before the fully connected layer, the standardization of the image input can be achieved. This layer is shown in Figure 4, and the specific implementation steps are as follows:

(1) The output of the convolutional layer is pooled several times, and the output of the pooling layer is improved;

(2) The normalized processing of the feature image is carried out after pooling;

(3) The obtained three feature images are cascaded;

(4) A feature image of the same size is obtained after the output.

#### 3.2.4. Fully Connected Layer

Each neuron in the fully connected layer is fully connected to the neuron in the previous layer. The fully connected layer can integrate local features in the convolutional layer or pooling layer, as given by the following:(3)$$x^{l}=\sum \alpha^{l-1}*W^{l}+b^{l},$$
where α represents the output of the previous layer, *W* represents the weight, and *b* represents the offset.

### 3.3. Classification

The softmax layer is selected as the classifier and final structural level for the FF-MICNN to realize the function of malicious code detection based on the extracted features. The neurons in the softmax layer are completely connected with the neurons in the upper layer. The activation function is the softmax function, which maps the result in the interval from 0 to 1. The mapped value is the probability of each category, and the probability of all categories adds up to 1.

The vector dimensions of the output of the softmax layer are determined by the number of types of malicious code datasets. In this detection model, the detection code results for which category. Therefore, the output of the softmax layer in this study is a nine-dimensional vector, which is given by the following equation:(4)$$f_{\theta }\left(x\right)=\frac{1}{e^{\theta_{0}x+\theta_{1}x+\cdots +\theta_{j}x}}\left[\begin{array}{c} \theta_{0}x\\ \cdots \\ \theta_{j}x \end{array}\right]=\left[\begin{array}{c} p(y=0|x,\theta )\\ \cdots \\ p(y=j|x,\theta ) \end{array}\right],$$
where θ represents the parameter matrix of the neural network, p(y|x,θ) represents the probability of category *Y*, namely the detection result, and the category with the largest value is taken as the target category.

## 4. Simulation and Analysis

### 4.1. Experimental Data

The data used in the Malicious Code Classification Competition initiated by Microsoft in 2015 were used in this study. These data contain nine different types of malicious code, including 10,867 samples that have been labeled. Table 4 presents the types and amounts of malicious code.

### 4.2. Evaluation Index

Table 5 presents the confusion matrix of the sample; TP, TN, FP, and FN can be calculated from the confusion matrix. TP represents the number of samples of a certain type of malicious code that is correctly classified as this type of malicious code after classification, TN represents the number of samples of other types of malicious codes that are correctly classified as other types of malicious codes after classification, FP represents the number of samples of certain types of malicious codes that are mistakenly classified as other types of malicious codes after classification, and FN represents the number of samples of other types of malicious codes that are mistakenly classified as these types of malicious codes after classification.

To facilitate and quantitatively analyze the detection effect of malicious code, unified evaluation indexes, including the accuracy rate, precision rate, recall rate, and F1 score, were used to evaluate the relevant performance of the model during the experiments conducted in this study. The calculation formulas for these four evaluation indexes are, respectively, as follows.
(5)$$accuracy=\frac{TP+TN}{TP+TN+FP+FN}$$
(6)$$precision=\frac{TP}{TP+TN}$$
(7)$$recall=\frac{TP}{TP+FN}$$
(8)$$F1=\frac{2*recall*precision}{recall+precision}$$

### 4.3. Parameter Setting

In the malicious code classification method based on FF-MICNN, FF-MICNN is used to automatically extract the deep features of malicious code. To show the comprehensive features of different malicious codes and improve the malicious code classification ability, the network structure of FF-MICNN was optimized via continuous experiments and parameter adjustment during the experiments. The parameters adjusted in the experiment included the learning rate and iteration number, among others. Only a single parameter was adjusted when all other parameters remained the same. The parameter with the best generalization power was selected and set as a fixed parameter, after which the next parameter was adjusted. This was continued until all parameters were adjusted and optimized, and an optimal parameter set of FF-MICNN was obtained. The model parameter setting is shown in Table 6.

### 4.4. Experimental Simulation

A deep learning framework was selected as the detection model in this study. After the improvement of the model, the limitation of converting malicious code files of different sizes into two-dimensional images is solved. The input criteria for the network model are determined by the fully connected layer. The neurons in the fully connected layer are fixed and are fully connected to the neurons in the previous layer. In this study, a layer was added before the fully connected layer to ensure the size of grayscale image features. Therefore, the improved network is referred to as the FF-MICNN. The following demonstrates the effectiveness of the network model by comparing the simulation results of single and fused features.

#### 4.4.1. Simulation Experiment of Opcode Sequence

As presented in Figure 5, the traditional machine learning algorithm, deep confidence network algorithm, and CNN algorithm were selected for analysis and comparison with the proposed FF-MICNN algorithm. The figure reveals that the detection ability of each model varied under different n-gram frequencies. However, from the overall view of the figure, the detection ability of the proposed model was relatively better than those of the other models. The detection ability of the proposed model was the best when the n-gram frequency of the opcode was about 700 and was about 0.2 percentage points higher than those of the other three models. Therefore, the opcode sequence detection of the deep learning model was effective.

#### 4.4.2. Grayscale Image Simulation Experiment

Figure 6 presents the simulation results of the effect of the use of grayscale image features on the FF-MICNN network model and other network models. It can be seen from the graph that the proposed network model was more stable than the other models, and its accuracy tended to be stable. On average, the accuracy of the proposed network model was higher than those of the other network models, and it was also more stable.

#### 4.4.3. Simulation Detection of Fused Features

To compare the influence of fused features and single features on model classification detection, opcode sequence features, grayscale image features, and fused features were respectively detected and compared to display the detection ability of single and fused features. The comparison is shown in Figure 7.

It can be seen from Figure 7 that the detection curve of fused features was always above the detection curve of a single feature. This indicates that feature fusion-based detection was better than single feature-based detection, and the accuracy reached 99.36%; this is 0.3% higher than the average accuracy of single feature-based detection, which is enough to demonstrate the superior detection ability of feature fusion.

The FF-MICNN was compared with the VGG15 network improved by VGG16 and a simple refinement of the CNN, as shown in Figure 8 and Figure 9. Although the performance of the VGG15 and CNN networks was better than that of the proposed network in approximately the first 15 rounds, with the increase in the number of training rounds, the FF-MICNN outperformed them. The reason for this is that a layer was added to the proposed method to solve the problem of unequal image sizes, and the appropriate number of convolution layers and the appropriate convolution kernel size were chosen. Regarding the other two networks, one is a simple 5-layer neural network, and the other lacks the processing of the image size. From the perspective of the network structure, the proposed network is more complex; thus, more specific image features can be extracted, thereby achieving a better detection effect.

To verify the detection capability of the model, three network algorithms, namely a parallel CNN, a CNN, and a CNN with a balanced dataset, were selected for comparison with the proposed FF-MICNN algorithm. The model was verified to have better performance in malicious code detection. The experimental results are reported in Table 7.

In Table 7, the four indexes of the true positive rate (TPR), false positive rate (FPR), accuracy, and F1 value are respectively compared and analyzed. This experiment resulted in good results on the performance ability of the proposed algorithm based on the four indexes. Among them, the accuracy index can best reflect the detection ability of the proposed model; the accuracy reached 98.6%, which is better than those of the other three detection algorithms. In terms of the TPR and FPR indexes, although the comparison of the accuracy index was not obvious, the results also indicate that the detection ability of this model was the best. Regarding the F1 score, the prediction accuracy of the proposed method was also good, indicating that the method has certain advantages in the aspect of malicious code feature extraction, thus leading to better detection.

In addition, the champion detection method of the Kaggle Competition and four methods presented in the extant literature [39,40,41,42] were also selected for comparative analysis to verify the effectiveness of the proposed feature selection method. The detection algorithm of the champion team uses the image features of malicious code, the fused features of opcode and headers, and the random forest algorithm to realize code detection. In the method by Lang et al. [39], three groups of features are extracted by texture maps and disassembly files for fusion classification, and the random forest algorithm is used as the classifier for code detection. In the method by Liu et al. [40], the frequency and behavior of the opcode sequence are used as feature vectors for dynamic and static feature fusion, and the KNN algorithm is then used to detect malicious code. In the method by Li et al. [41], grayscale image and color feature vectors are fused, and malicious code detection is then realized via the random forest algorithm. In the method by Luo et al. [42], texture image features and command frequency features are used for feature fusion, and a deep confidence network is used for detection. These five algorithms were compared with the proposed FF-MICNN algorithm, and the results show that the FF-MICNN model achieved better performance in malicious code detection. The experimental results are presented in Table 8.

According to the analysis method described previously, two characteristic features were selected for use in the proposed method, and very high detection accuracy was achieved. It was verified that the two features selected for use in the proposed method can sufficiently express the global and local features of malicious code, and almost all the features of malicious code can be extracted for detection. In addition, the proposed method can automatically learn and extract deep-level features; this is different from the machine learning algorithm, which extracts only surface-level features.

The fusion of the two features extracted from the FF-MICNN model can achieve a comprehensive representation of malicious code features, which can improve the classification detection of the model to a certain extent. In addition, the fused features can reveal more superficial features than single features in the representation of malicious code features, and deeper features can be further obtained through the FF-MICNN model, thus improving the results of malicious code detection. Therefore, the proposed model has a certain significance for the detection of malicious samples. It not only achieves detection with less time and resource consumption but also is not subject to the amount and type of malicious code, and can solve the problems of the explosive growth of malicious code and analysis difficulty.

In this work, a malicious code detection method based on the combination of the sequence features of the opcodes and grayscale image features was proposed. This algorithm mainly extracts the features of the sequence of operation codes as the local features and extracts the features of malicious code disassembly files converted into grayscale images as the global features. The two features are fused through a fusion algorithm and input into the FF-MICNN for comprehensive abstract feature extraction. Classifiers are used to divide them into respective categories to complete the detection of malicious code. The experimental results show that feature fusion is more conducive to the representation of code features and can represent more comprehensive and abstract features, which aids in the detection of malicious code.

## 5. Conclusions

This paper proposed a malicious code detection algorithm to solve the problems of the constant increase of malicious code and deformation based on malicious code. First, the confused code causes the expression of extracted features to be unclear, and its behavior and instructions are difficult to understand. Therefore, the algorithm can avoid the confusion of malicious code by converting the code into grayscale image space vectors, and the detection of malicious code can be converted into an image detection task. Second, the grayscale image input into the CNN limits the size of the grayscale image, so the extracted feature expression is incomplete. Hence, the CNN is improved by adding a structural layer to realize the function of the input of grayscale images without limiting the size, thereby improving the detection performance. The results of simulations demonstrated that this algorithm is better than the machine learning algorithm in terms of the objective indicators of the accuracy, recall rate, and precision rate, and more accurate detection results were obtained.

Aiming at the problem of the incomplete feature description of malicious code, a feature fusion detection algorithm was proposed. First, the opcode sequence in the disassembled .asm file of malicious code is selected as the local feature, and the .asm file is then transformed into a grayscale image space vector as the global feature. The static features of the malicious code can be described completely by fusing these two features. The fused features are input into the FF-MICNN to realize automatic feature extraction, and the types of malicious code are represented by the softmax layer in the form of probability, thus realizing the detection of malicious code. The results of simulations revealed that the proposed algorithm can fully express the characteristics of malicious code because of the fusion of local and global features. The proposed FF-MICNN realizes the deep feature extraction and autonomous learning of malicious code and avoids human interference. The results of the simulations showed that the accuracy of the proposed algorithm was better than those of algorithms using a single feature.

Although the proposed algorithm is optimized for the graphical processing of malicious code, it is characterized by some shortcomings. In the present study, the test dataset was unbalanced and the accuracy was meaningful in the specified condition. Furthermore, the test dataset size was not large, which cannot indicate the superiority of FF-MICNN. Moreover, the algorithm does not take into account the uneven distribution of malicious code types in the dataset. In future research, the test dataset will be balanced and enlarged to further improve the proposed malicious code detection algorithm under these conditions.

## Figures and Tables

**Figure 1 sensors-22-08739-f001:**
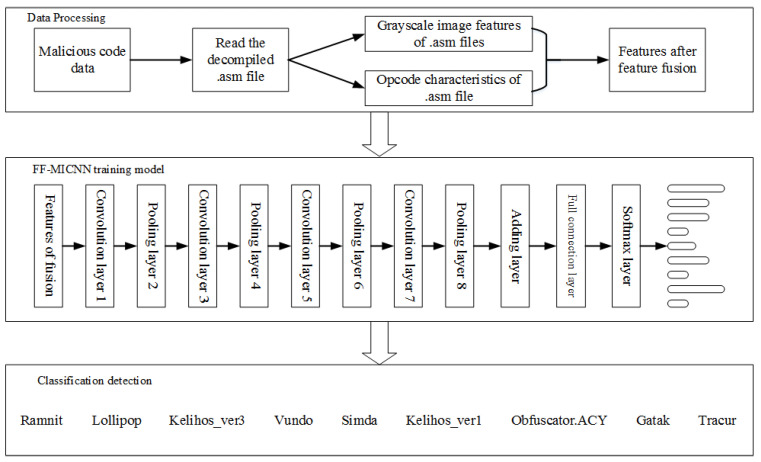
Malicious code detection method based on FF-MICNN.

**Figure 2 sensors-22-08739-f002:**
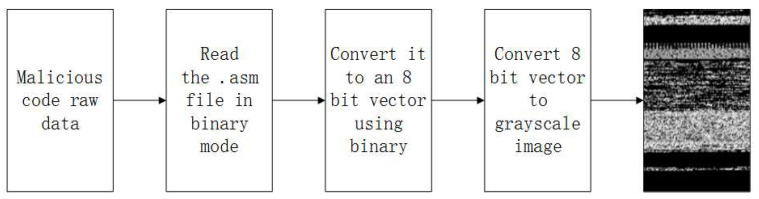
Malicious code visualization process.

**Figure 3 sensors-22-08739-f003:**
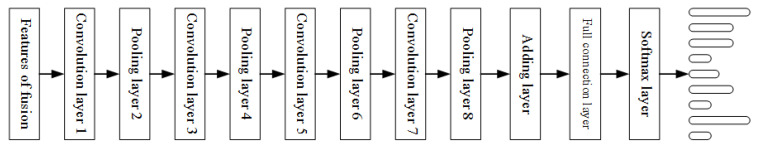
Network structure of malicious code detection based on FF-MICNN.

**Figure 4 sensors-22-08739-f004:**
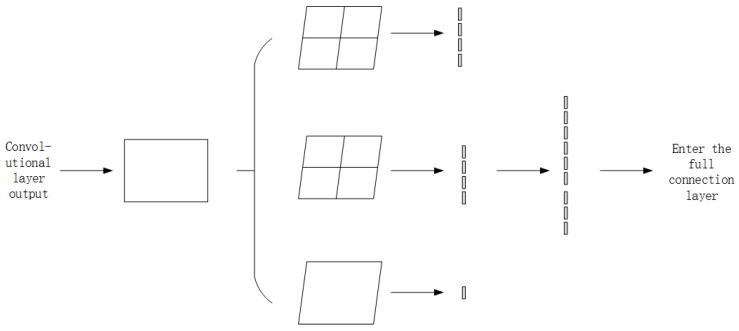
Process of adding layers.

**Figure 5 sensors-22-08739-f005:**
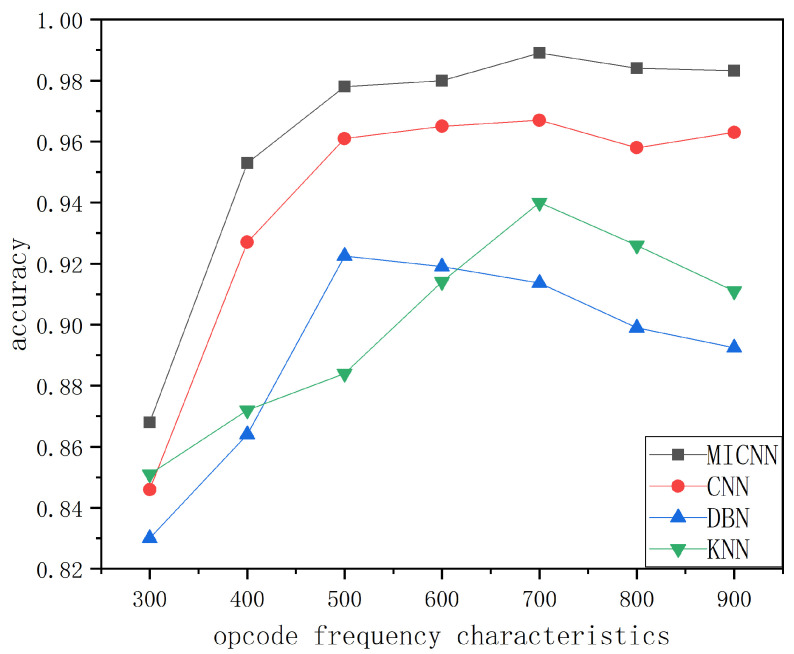
Simulation of opcode sequence.

**Figure 6 sensors-22-08739-f006:**
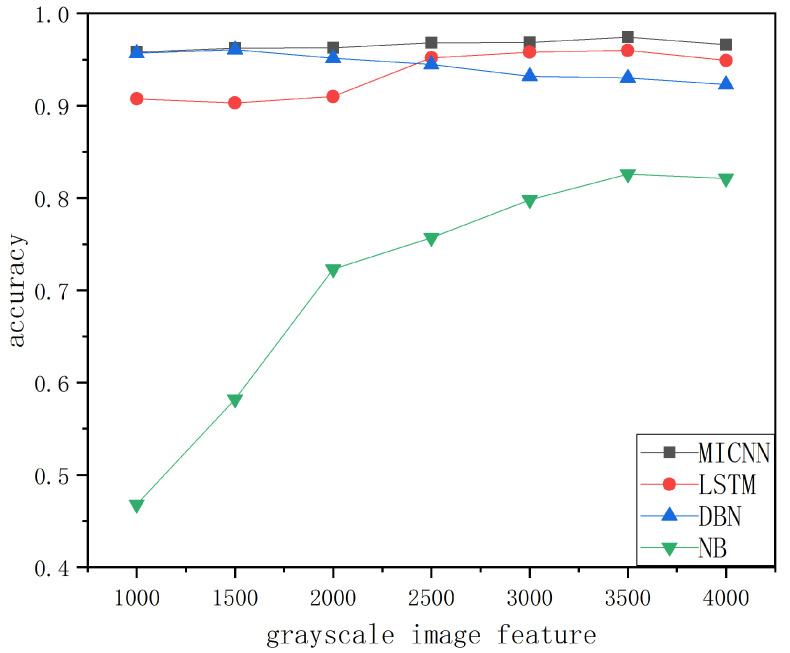
Simulation of gray image.

**Figure 7 sensors-22-08739-f007:**
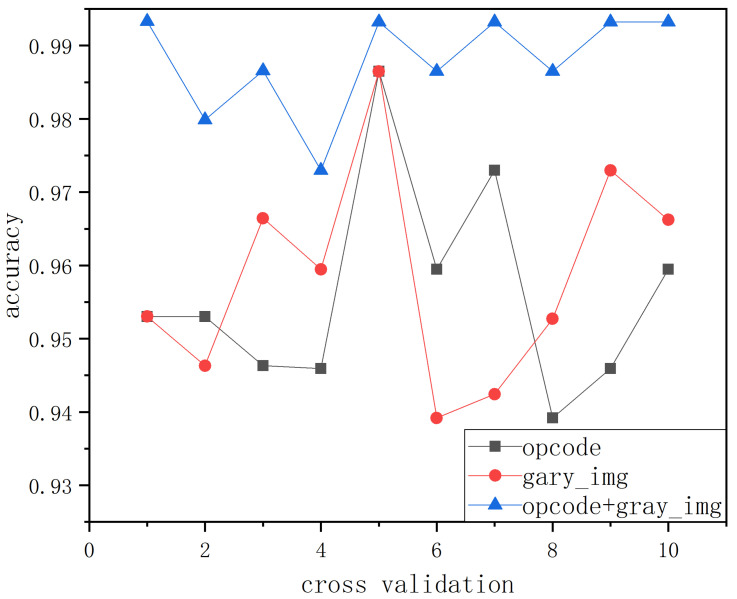
Simulation of single feature and fusion feature.

**Figure 8 sensors-22-08739-f008:**
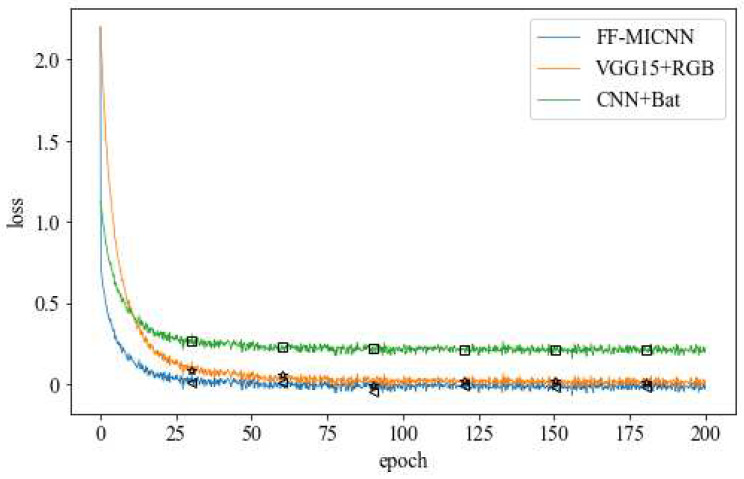
Change of loss in network training.

**Figure 9 sensors-22-08739-f009:**
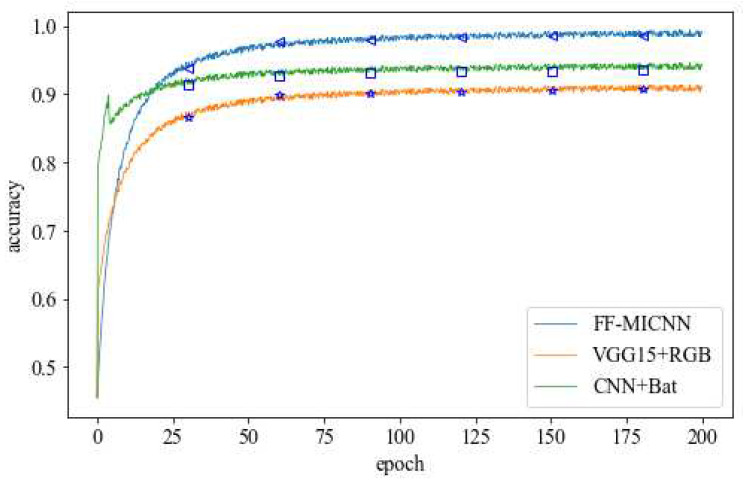
Change of accuracy in network training.

**Table 1 sensors-22-08739-t001:** The static detection methods.

Authors	Algorithm Description	Merits
Zhu et al. [10]	A malicious code detection method using	Can quickly detect malicious code,
	API sequence of malicious code.	cannot detect newly emerged
		malicious code.
Zhang et al. [11]	Use attribute similarity to detect	Can quickly analyze malicious
	malicious code.	code, cannot accurately analyze
		confused code.
Abhijit et al. [12]	A malicious code detection method based	Has high accuracy and can detect
	on frequency characteristics.	wrong and missed code, consumes a
		large number of resources.
Kang et al. [13]	An n-gram opcode features-based approach	Supports automatic feature discovery
	that utilizes machine learning to identify	without relying on prior experts
	and categorize Android malware.	or domain knowledge.
Imran et al. [14]	A detection scheme based on the hidden	Relies heavily on the API sequence,
	Markov model and discriminant classifier.	and requires a large amount of
		computation.
Siddiquiet et al. [15]	A worm prevention technology using a	Can improve the detection rate of
	data mining framework.	new worms without using a large
		amount of data.
Moser et al. [16]	A binary obfuscation scheme.	Easily avoided if malicious code is
		packaged or confused.

**Table 2 sensors-22-08739-t002:** The dynamic detection methods.

Authors	Algorithm Description	Merits
Hisham et al. [17]	A behavior-based feature model can	Can quickly detect malicious code,
	dynamically analyze and evaluate	cannot detect newly emerged
	a dataset of malicious code.	malicious code.
Li et al. [18]	A detection method based on the	Malicious code can be detected,
	semantic dynamic characteristics of	evasive malicious code cannot.
	malicious code.	
Rong et al. [19]	A MACSPD detection method, which is	Better than a similar type of algorithm
	an API sequence model mining method.	in detecting unknown malicious code.
Ucci et al. [20]	A scalable clustering method to	Can identify and group similar malware
	identify and group malware samples.	programs with better accuracy.
Phodeet et al. [21]	A model that predicts malware in	Determine the presence of malware
	executing files.	before it executes a payload.
Mohaisenet et al. [22]	A behavior-based automated malware	Difficult to realize this method in the
	analysis and classification System.	case of too much malicious code data.

**Table 3 sensors-22-08739-t003:** The Image detection methods.

Authors	Algorithm Description	Merits
Wan et al. [24]	A malicious code classification	Can improve the performance of the
	method based on analytic behavior.	algorithm while reducing data dimensions.
Han et al. [25]	A malicious code detection method	Cannot sufficiently solve the
	based on the texture features of	artificial influence, nor can it
	malicious code.	achieve end-to-end detection.
Hashemi et al. [26]	An image-based method to detect	Can be regarded as a framework with
	unknown malicious code.	flexibility.
Xiao et al. [27]	A strategy to select a deep learning	Classification accuracy was 99.72%
	model that fits the malware	higher than that of other
	visualization images.	classification methods.

**Table 4 sensors-22-08739-t004:** The types of malicious code.

Serial Number	Malicious Code Category	Description	Sample Size
1	Lollipo	—	2478
2	Ramni	Contains the code for a powerful botnet	1542
3	Simda	Consists of four types of malicious code,	
		The most sophisticated of which are botnets,	42
		Trojans, backdoors, and password theft	
4	Vundo	Trojans and worms	474
5	Tracur	Trojans	751
6	Gatak	Trojan horse	1013
7	Kelihos_ver3	Encrypted P2P botnets	2942
8	Kelihos_ver1	Bot	398
9	Obfuscator.ACY	Malicious code formed by a	
		combination of four methods	1228

**Table 5 sensors-22-08739-t005:** The confusion matrix.

Types	Real Samples	Pseudo-Samples
Real samples	TP	FN
Pseudo-samples	FP	TN

**Table 6 sensors-22-08739-t006:** The parameter setting of the model network.

Parameter	Numerical Value	Instruction
Input size	variable	Fusion features of malicious code
Convolution kernel	3 × 3	Size of the convolution kernel
Step length	1	Step size of the convolution kernel and pooling window
Pool size	2 × 2	Size of the pool window
Learning rate	0.001	Learning rate of FF-MICNN
Iterations	15	Number of iterations
Activation function	ReLU	Activation function selected by FF-MICNN
Classifier	softmax	Softmax regression classification model
Dataset partition	7:3	Ratio of training data to test data

**Table 7 sensors-22-08739-t007:** The comparison of different methods.

Algorithm Name	TPR	FPR	Accuracy	F1
Parallel-CNN [30]	0.959	0.0326	0.9818	0.9805
CNN-Image [31]	0.9387	0.0735	0.9406	0.9317
CNN+Image+Bat [32]	0.9255	0.0311	0.9346	0.9221
FF-MICNN	0.9614	0.0311	0.986	0.9817

**Table 8 sensors-22-08739-t008:** Comparison of different methods.

Paper Author	Accuracy	Description
Kaggle champion	98%	Feature fusion of image features, opcodes, and headers
Lang et al.	87%	Feature fusion of opcode sequences and gist features
Xiu et al.	96%	Feature fusion of opcode sequence frequency and behavior
Li et al.	95.518%	Feature fusion of grayscale image and color
Luo et al.	95.76%	Feature fusion of grayscale image texture and operation frequency
FF-MICNN	98.6%	Truncated scale grayscale image and opcode sequence

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
