# Peer review of "A Malicious Code Detection Method Based on FF-MICNN in the Internet of Things"

_sensors, 2022, doi:10.3390/s22228739_

Round 1

Reviewer 1 Report

Review of A Malicious Code Detection Method Based on FF-MICNN in the Internet of Things

This paper address an interested problem of detect malicious code. The paper proposed a method that transform code into gray scale image, and then using convolution layers extract features from the image finally through a dense layer to detect malicious code.

The paper give a good review of current research in malicious code detection. The authors also compared the proposed method with other ones.

However, the paper is not well written. Sentences are repeated.

The experiment is not describe what is so-called “Pseudo samples” and what size of the Pseudo samples”. The comparison use F1 shows the proposed algorithm only litter better (0.012) than Parallel-CNN. If the test size is not big enough, this cannot indicate FF-MICNN is better.

The paper also use accuracy to do comparison. But if the test dataset is unbalanced compare accuracy may meaningless.

Author Response

Added in file.

Reviewer 2 Report

1. A lot of grammatical mistakes. The authors of this manuscript are suggested to proofread this article.

2. All references are not latest. The authors are suggested to include some references from 2022.

3. The abstract indicates only a qualitative assessment of the overall performance while a quantitative assessment is expected.

4. Related works should better be tabulated showing the analysis of the author of the present state.

5. What is the source of equation 1, 2, 3, 4 etc. ?

6. Explain, what is x and f in equation 1.

7.  Include the motivation of the research in this manuscript.

8. Include all texonomy in a table.

9. Discuss the findings in a seprate section before conclusion.

10. Future scope should be specified.

Author Response

Added in file

Round 2

Reviewer 2 Report

The authors have revised this manuscript according to my previous comments. I am agree to accept this manuscript in its present form.

Author Response

Many thanks for the reviewer’s valuable comment. we have carefully checked our manuscript and modified the mistakes.
